# Release of *Plasmodium* sporozoites requires proteins with histone-fold dimerization domains

Chiara Currà[1], Renate Gessmann[1], Tomasino Pace[2], Leonardo Picci[2], Giulia Peruzzi[3], Vassiliki Varamogianni-Mamatsi[1], Lefteris Spanos[1], Célia R.S. Garcia[4], Roberta Spaccapelo[3], Marta Ponzi[2] & Inga Siden-Kiamos[1]

The sporozoite, the stage of the malaria parasite transmitted by the mosquito, first develops for ∼2 weeks in an oocyst. Rupture of the oocyst capsule is required for release of sporozoites, which then transfer to the salivary gland where they are injected into a new host. Here we identify two parasite proteins that we call oocyst rupture proteins 1 (ORP1) and ORP2. These proteins have a histone-fold domain (HFD) that promotes heterodimer formation in the oocyst capsule at the time of rupture. Oocyst rupture is prevented in mutants lacking either protein. Mutational analysis confirms the HFD as essential for ORP1 and ORP2 function, and heterodimer formation was verified *in vitro*. These two proteins are potential targets for blocking transmission of the parasite in the mosquito.

[1] Foundation for Research and Technology—Hellas, Institute of Molecular Biology and Biotechnology, N. Plastira 100, GR 700 13 Heraklion, Greece. [2] Istituto Superiore di Sanità, Dipartimento di Malattie Infettive, Parassitarie ed Immunomediate, 0161 Roma, Italy. [3] Department of Experimental Medicine, University of Perugia, 06132 Perugia, Italy. [4] Departamento de Fisiologia, Instituto de Biociências, Universidade de São Paulo, São Paulo 05508-900, Brazil. Correspondence and requests for materials should be addressed to I.S.-K. (email: inga@imbb.forth.gr).

The oocyst of the *Plasmodium* parasite, the causative agent of malaria, is the longest life stage of the parasite. The oocyst is formed about 24 h after the uptake of a blood meal by the mosquito. During these first hours, the sexual stage of the parasite results in the formation of the elongated motile ookinete, which traverses the midgut epithelium and comes to rest under the basal lamina. Here, the ookinete undergoes radical changes in cellular architecture, resulting in the formation of the round oocyst, which is separated from the insect by a little known structure called the oocyst capsule. Concomitant with oocyst growth DNA is replicated and ~13 nuclear divisions take place within a syncytium. Retraction of the oocyst plasma membrane from the capsule results in cytoplasmic islands, called sporoblasts, from which thousands of sporozoites bud off into the space delineated by the capsule. Finally, the capsule ruptures to release the sporozoites into the haemocoel. The rupture is dependent on a parasite-derived cysteine protease called ECP1 (ref. 1) and also on a repeated region of the circumsporozoite protein (CSP)[2], a constituent of the oocyst plasma membrane and a prominent sporozoite surface protein.

The histone-fold domain (HFD) is found in histones and in proteins with a role in transcriptional regulation such as the TATA-box-binding protein-associated factor $TAF_{II}$ and in the CCAAT-binding transcription factor subunits NF-YB and NF-YC[3]. The only example of a protein with a HFD acting outside the nucleus is son-of-sevenless, a protein with multiple domains containing two HFDs, which are involved in binding to lipids[4]. The HFD is ~70 amino acids in length and forms three helices separated by small linker sequences. In a heterodimer the proteins are organized in head-to-tail orientation, resulting in a compact 'handshake' interaction[5]. In the well-studied NF-Y complex, the heterodimer NF-YB and NF-YC interacts with a third subunit, NF-YA[6]. Although the heterodimer binds DNA in a nonspecific manner, the NF-YA subunit confers binding specificity to the CCAAT motif[7].

Here we show that each of two HFD-containing proteins of the *Plasmodium* parasite (*Plasmodium berghei* gene IDs PBANKA_0902500 and PBANKA_1303400), which we name oocyst rupture protein 1 (ORP1) and ORP2, have essential and similar functions in the rupture of the oocyst capsule. We provide evidence that protein interaction via the HFD is critical for function. Furthermore, ORP1 is located in the oocyst capsule, whereas ORP2 is re-localized from the oocyst cytoplasm to the capsule at the time when mature sporozoites have formed. Taken together, these findings allows us to hypothesize that interaction of the two ORP proteins in the capsule, directly or indirectly, leads to a destabilization of this structure allowing the sporozoites to escape into the mosquito haemocoel.

## Results

**Two *Plasmodium* HFD proteins are essential for oocyst rupture.** The malaria parasite genome encodes two HFD-containing proteins, which we in the following call ORP1 and ORP2. ORP1 (PBANKA_0902500, 950 amino acids) contains a carboxy-terminal HFD. BLAST searches revealed the protein to be most similar to NF-YB of plants and animals, and the HAP3 transcription factor of yeast (Fig. 1a,b). The protein has been detected previously in *Plasmodium falciparum* asexual blood stages and its expression was suggested to be modulated by melatonin[8]. The *orp2* gene product (PBANKA_1303400, 875 amino acids) has an amino-terminal HFD, which is most similar to NF-YC of plants and animals, and HAP5 of yeast (Fig. 1a,c). Both HFDs comprises the three α-helices characteristic of the HFD and the short αC helix found in NF-YB/NF-YC proteins. Neither of the two proteins contains any other recognizable motifs and outside the HFD there is only a low degree of

similarity comparing the two (Fig. 1a–c and Supplementary Fig. 1). The two proteins are considerably longer than NF-YB and NF-YC of animals. BLAST searches of the *Plasmodium* genome failed to reveal any protein with similarity to NF-YA, which suggested that these two proteins may not be part of a classical NF-Y DNA-binding complex. We thus hypothesized that these two proteins carry out divergent function(s) in the malaria parasite.

To investigate the functions we produced independent gene disruption mutants of *orp1* and *orp2* in *P. berghei*, named *orp1(−)* and *orp2(−)*, respectively (Supplementary Fig. 2). Two clones of each mutant were studied. Asexual growth was not different from the wild type (WT) (Supplementary Fig. 3) and sexual development resulted in formation of ookinetes in similar numbers to the WT (Supplementary Table 1). Mosquitoes were infected with each line and at day 11 post blood feeding (p.b.f.) oocysts of normal size were detected (Fig. 1d, Supplementary Fig. 4a and Supplementary Table 1). The average numbers of oocysts formed in the two mutants were higher at day 11 p.b.f. than in WT infected mosquitoes and this was especially prominent in *orp1(−)*. The significance of this was not further investigated.

In WT infected mosquitoes the oocysts rupture at ~12–14 days p.b.f. and sporozoites are released, which then travel to the salivary gland where they can be detected at 18–21 days. In our two mutants, although the oocysts developed normally and sporozoites were formed, they did not rupture and persisted in the midguts at least until day 21 (Fig. 1e,f, Supplementary Fig. 4b and Supplementary Table 1). To determine whether the sporozoites were motile, oocysts were mechanically disrupted and the sporozoites were allowed to glide on glass slides. Typical circular trails were visualized after labelling with an antibody against the secreted protein CSP[9], indicating that the sporozoites were motile (Supplementary Fig. 5a and Supplementary Methods). Consistent with impaired oocyst rupture, in mosquitoes fed with the two mutants no sporozoites were detected in the salivary gland at day 18 or later p.b.f. (Table 1 and Supplementary Table 1). Conversely, the mutant oocysts contained several thousand sporozoites when counting them from homogenized midguts at this time point (Table 1 and Supplementary Table 1). Infected mosquitos were allowed to bite directly on naive mice 21 days p.b.f., but in no case were the mutants able to establish an infection, whereas WT controls were positive after 3 days (Table 2). We next investigated whether the *orp1(−)* and *orp2(−)* mutant sporozoites isolated from midguts 18 days p.b.f. were able to infect mice after injection in the tail vein. In this case as well, the mutants failed to establish an infection (Table 2). This suggested functions of these proteins in the ability of sporozoites to establish an infection in the animal. Attempts to detect the mutants 44 h post infection in the mice by reverse transcriptase–PCR were negative (Supplementary Fig. 5b and Supplementary Methods), suggesting an early block of mutant parasite development. Further investigations are thus necessary to determine at which stage these mutants are arrested in the liver stage.

**ORP1 localizes to the oocyst capsule.** To investigate this unexpected phenotype further, we determined the localization of ORP1 using a specific antibody developed against an eight amino acid peptide in the *P. falciparum* orthologue of ORP1 (ref. 8), which differs in one amino acid from *P. berghei* (Fig. 1b, green box, validation of the antibody is shown in Supplementary Fig. 6 and Supplementary Methods). We also generated a *P. berghei* line expressing the ORP1 N-terminal 365 amino acids fused to green fluorescent protein (GFP) (Fig. 2b and Supplementary Fig. 7a). Both approaches gave consistent results. In ookinetes and

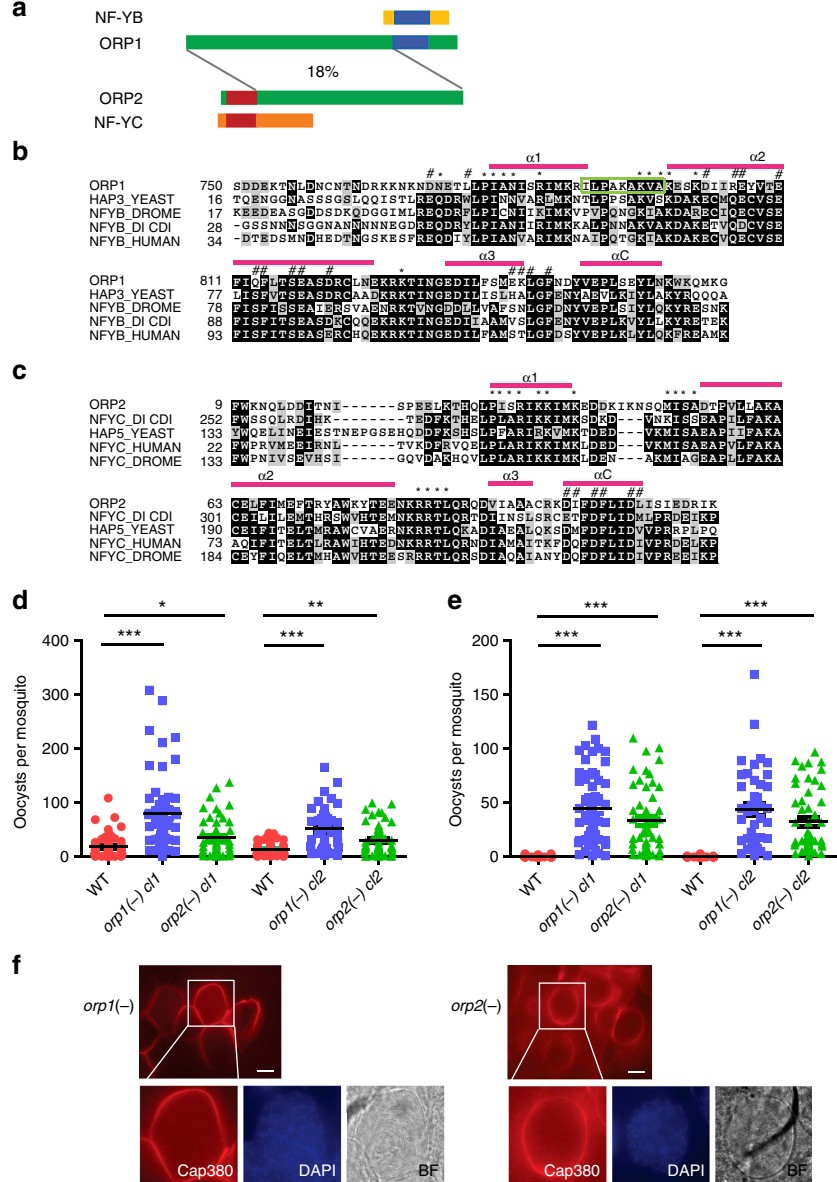

**Figure 1 | NF-YB/NF-YC proteins are essential for oocyst rupture.** (**a**) Schematic representation of the HFD containing proteins in *P. berghei* (ORP1, PBANKA_0902500 and ORP2, PBANKA_1303400) compared with the human NF-YB and NF-YC. HFD is indicated in blue in the upper part and in red in the bottom part. (**b**) Multiple sequence alignment of N-terminal portion of NFY-B from human (aa 34–146), *Drosophila melanogaster* (aa 17–130) and *Dictyostelium discoideum* (aa 28–140), HAP3 of *Saccharomyces cerevisae* (aa 16–129) and aa 750–863 of ORP1. The peptide used for antibody generation is marked with a green rectangle. Accession numbers: P13434.1HAP3_YEAST, Q54WV0.1NFYB_DICDI, P63140.1, P25208.2 NFYB_HUMAN and NP_609997.1 (*D. melanogaster*). (**c**) Multiple sequence alignment of HFD of NFY-C from human (aa 22–125), *D. melanogaster* (aa 133–236) and *D. discoideum* (aa 252–353), HAP5 of *S. cerevisae* (aa 133–236) and aa 9–115 of ORP2. Accession numbers: Q557I1.1 NFYC_DICDI, NP_572354.1 (*D. melanogaster*), Q13952.3 NFYC_HUMAN, Q02516.1 HAP5_YEAST. In **b,c**, the α-helices of the HFD (α1–α3) are denoted in red on top of the sequence alignment and also the NF-Y-specific αC domain. The domains are derived from a published alignment[7]. (**d,e**) Oocyst loads of WT (red circles), *orp1(−) cl1* and *cl2* (blue squares), and *orp2(−) cl1* and *cl2* (green triangles) mutants. The data represent pooled data from two experiments of each clone. The complete data set is presented in Supplementary Fig. 4 and Supplementary Table 1. ***$P < 0.0001$, **$P < 0.01$ and *$P < 0.05$, Mann–Whitney test. Error bars denote s.e.m. (**d**) Oocyst loads counted at day 11 p.b.f. (**e**) Oocyst loads counted at day 21 p.b.f. (**f**) Representative pictures of *orp1(−) cl1* (left) and *orp2(−) cl 1* (right) oocysts containing sporozoites in midguts dissected at day 20 p.b.f. The oocysts were labelled with Cap380 (red). The magnification of one oocyst shows also the nuclei of the sporozoites (4,6-diamidino-2-phenylindole (DAPI), blue) and the bright field image (BF) sporozoites within the oocysts. Scale bar, 15 μm. WT infected mosquitoes contained very few degenerated oocysts and are not shown.

sporozoites (Fig. 2a,c) the protein was detected in the cytoplasm but not in the nucleus, whereas in the oocyst stage ORP1 was found in the periphery of the mature cyst (Fig. 2a). Double-labelling experiments revealed that ORP1::GFP co-localizes with the protein Cap380 (ref. 10), the only protein of the oocyst capsule described until now, supported also by the presence of

ORP1 outside the cell membrane highlighted with the antibody against the CSP[11–14] (Fig. 2c,d). We thus propose that ORP1 is a capsule protein. It is the second parasite protein that has been identified in the capsule, the composition of which is not well understood. The localization of ORP1 to the oocyst periphery was independent on the presence of ORP2 (Fig. 2e).

| Table 1 | Midgut and salivary gland sporozoites. | | | |
|---|---|---|---|---|
| **Experiment** | **Parasite** | **Midgut sporozoites** | **Salivary gland sporozoites** | **Number of mosquitoes dissected** |
| #1 | WT | 0 | 4,050 | 61 |
| | orp1( − ) cl1 | 4,600 | 0 | 59 |
| | orp2( − ) cl1 | 5,900 | 0 | 46 |
| | hfd− cl 1 | 3,450 | 0 | 45 |
| #2 | WT | 0 | 4,100 | 51 |
| | orp1( − ) cl2 | 4,400 | 0 | 43 |
| | orp2( − ) cl2 | 4,050 | 0 | 49 |
| | hfd− cl 2 | 2,700 | 0 | 45 |
| #3 | WT | 22,917 | 25,900 | 240 |
| | orp1( − ) cl2 | 24,885 | 0 | 217 |
| | orp2( − ) cl2 | 16,188 | 0 | 244 |

In experiment 1 and 2, midguts and salivary glands were dissected at day 21 p.b.f. In each experiment, the data are pooled from two experiments of each clone. The complete data set is presented in Supplementary Table 1. In experiment 3, midgut and salivary glands were dissected from the same mosquito at day 18 p.f.b. In all experiments, the organs were homogenized and sporozoites counted under the microscope.

| Table 2 | Infectivity of sporozoites in mice. | | |
|---|---|---|---|
| **Experiment** | **Parasite** | **Pre-patent*** | **Number of infected mice** |
| #1 Mosquito bite | WT | 3.0 | 3/3 |
| | orp1( − )cl1 | 0 | 0/3 |
| #2 Mosquito bite | WT | 3.0 | 3/3 |
| | orp2( − )cl1 | 0 | 0/3 |
| #3 Mosquito bite | WT | 3.0 | 3/3 |
| | orp1hfd− cl1 | 0 | 0/3 |
| #4 Intravenous injection | WT | 6.5 | 8/8 |
| | orp1( − )cl1 | 0 | 0/8 |
| | orp2( − )cl1 | 0 | 0/8 |
| #5 Intravenous injection | WT | 5.5 | 8/8 |
| | orp1( − )cl2 | 0 | 0/8 |
| | orp2( − )cl2 | 0 | 0/8 |

Experiment 1 to 3: 20 infected mosquitoes were allowed to feed on naive C57BL/6 mice.
Experiment 4 and 5: $5 \times 10^5$ sporozoites isolated from the midguts of infected mosquitoes 18 days p.b.f. were injected intravenously in the tail vein of C57BL/6 mice.
*Pre-patent: the time between sporozoite inoculation and the appearance of parasites in the blood (days).

**ORP2 is re-localized to the capsule just before rupture**. To determine the localization of ORP2, a line was created expressing a C-terminal fusion of the endogenous gene to mCherry (ORP2::mCherry) (Fig. 3a and Supplementary Fig. 7b). In this line, no mCherry signal was detected in asexual stages, gametocytes, ookinetes or sporozoites after examining the cells in immunofluorescence assays (IFA). However, in oocysts it was readily detected in the cytoplasm in oocysts at day 7 (Fig. 3b). Cytoplasmic localization was maintained in oocysts until day 12 and before sporozoite formation (Fig. 3c). When sporozoites had formed and were ready to egress from the oocyst, ORP2 was re-localized to the oocyst periphery where it partly co-localized with Cap380 (Fig. 3d).

**ORP1 and ORP2 HFDs form a dimer *in vitro*.** Our results thus identified the two HFD proteins as necessary for the timely rupture of the capsule to release mature sporozoites. Based on their common function and peripheral localization in mature oocysts ready to burst, we reasoned that these proteins may dimerize through the NF-YB and NF-YC domains. Structural modelling showed that the two parasite HFD's could form a dimer with a very similar structure to that of the NF-YB/NF-YC dimer (Fig. 4a and Supplementary Movie 1). To verify whether the two HFDs are indeed able to form a dimer, we expressed the two HFDs in *Escherichia coli* (for details see Methods). We used the pET-Duet-1 vector, which allows co-expression of the two proteins in the same cell. ORP1 was expressed with a C-terminal *S*-tag (calculated molecular weight 18,38 kDa) and ORP2 with an N-terminal poly-His tag (calculated molecular weight 19,91 kDa). Both proteins were expressed as soluble peptides at the expected size, with a lower band of each peptide also recognized by the antibodies, presumably processed forms (Supplementary Fig. 8a). A Ni-NTA agarose pull-down assay demonstrated that the *S*-tagged ORP1 peptide was retained on the resin together with the poly-His-tagged ORP2, thus supporting that the two domains can form a complex (Fig. 4b). As a control, ORP1 was expressed from the same vector but without the ORP2 protein. In this case, no binding to the Ni-NTA agarose was observed (Supplementary Fig 8b). We also performed a co-immunoprecipitation experiment. The antibody against the *S*-tag was bound to Protein G Sepharose beads and incubated with the cleared lysate containing the two peptides from *E. coli*. We readily identified His-ORP2 on western blottings loaded with the material captured on the beads (Fig. 4c). This provides additional evidence that the two peptides interact.

**The HFD is essential for ORP1 function**. Next, we investigated whether the dimerization domain was essential for the rupture of the oocyst. We produced a mutant called *orp1-hfd−* in which the ORP1 protein was lacking part of the α2 to αC domains of the HFD (amino acids 797–860) but leaving the epitope for the antibody intact (Fig. 5a and Supplementary Fig. 2e,f). The *orp1-hfd−* mutants formed similar number of oocysts to WT (Fig. 5b and Supplementary Fig. 9a). The mutant protein still maintained a peripheral localization (Fig. 5c) consistent with the localization of the GFP fusion of the C-terminally truncated ORP1 to the capsule (Fig. 2c), which thus confirms that localization is due to motifs in the N-terminal part of the protein and is independent of the dimerization domain. Importantly, in mosquitoes infected with *orp1-hfd−*, oocysts were formed but did not rupture even after 21 days (Fig. 5b,c and Supplementary Fig. 9b). Consequently, no sporozoites were detected in salivary glands of the *orp1-hfd−* infected mosquitoes (Supplementary Table 1) and there was no transmission to naive mice by bite (Table 2). These results suggest that the ORP1-HFD is essential for the function of the protein in oocyst rupture.

## Discussion

In this study we show that rupture of oocysts is strictly dependent on the parasite proteins ORP1 and ORP2, each containing an HFD with similarity to the NF-YB and NF-YC subunits of the NF-Y transcription factor. ORP1 is located in the oocyst capsule from an early time point, whereas ORP2 is transiently trafficked from the cytoplasm to the capsule when mature sporozoites have formed. Our results also provide evidence that the two proteins form a dimer via the HFD. A mutant where the C-terminal 63 aa of the ORP1 HFD was deleted had the same phenotype as the deletion of the complete *orp1* coding region. Furthermore, the two HFD of the ORPs form a dimer *in vitro* as demonstrated by pull-down assay and co-immunoprecipitation. Based on these data, we propose that rupture of the oocyst capsule is regulated by the dimer formation of ORP1 and ORP2 just-in-time when

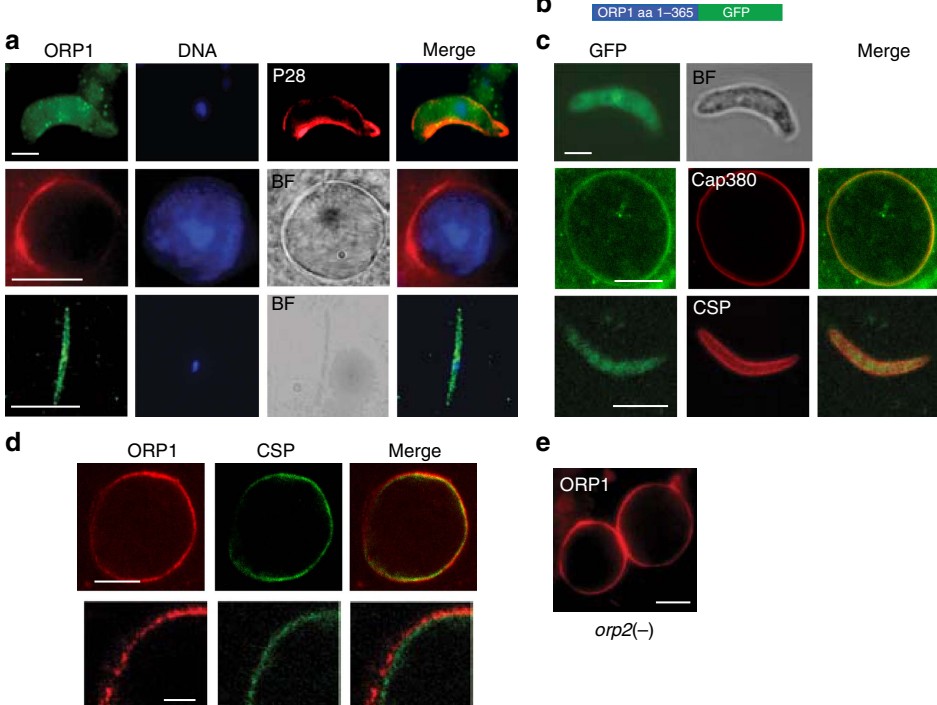

**Figure 2 | ORP1 is expressed in the oocyst capsule and localization is independent of ORP2.** (**a**) Localization of ORP1 using an antibody directed against a peptide of the HFD in ookinetes, oocysts at 10 days p.b.f. and sporozoites. Scale bars, 2.5, 15 and 10 μm, respectively. The ookinete was also labelled with an antibody directed against the P28 surface protein (red). Nuclei were stained with 4,6-diamidino-2-phenylindole (DAPI; blue). BF, bright field image. (**b**) Schematic depiction of a construct encoding a GFP fusion of the N-terminal 365 aa fragment of ORP1. (**c**) Localization of ORP1::GFP in ookinetes, oocysts at 6 days p.b.f. and sporozoites using immunolabelling. Of note is the localization of ORP1::GFP in proximity to Cap380 in the oocyst periphery. Scale bars, 2.5, 10 and 5 μm, respectively. (**d**) ORP1 is localized to the oocyst capsule revealed by double labelling with antibodies recognizing ORP1 and CSP, at day 7 bpf (scale bars, 10 μm (top row) and 2.5 μm (bottom row)). (**e**) Immunolabelling of an orp2( − ) mutant oocyst reveals localization of ORP1 (red) at the oocyst periphery. Scale bar, 15 μm.

mature sporozoites have been formed. This may directly or indirectly trigger a change in the capsule, leading to its destabilization. One possibility is that the dimer leads to the activation of the ECP protease, which has been shown to be essential for rupture of the capsule[1]. Alternatively, the dimer may lead to structural changes in target proteins of this protease, making them susceptible to protease cleavage.

The HFD is a protein domain involved in heterodimerization and is found in many DNA-binding classes of proteins. The NF-Y transcription factor consists of three subunits, A, B and C, with the B and C subunits containing each an HFD. Structural and biochemical data have shown that the NF-YB and -YC subunits first form a dimer via the HFD, which can bind DNA nonspecifically[7]. The addition of the NF-YA subunit confers specificity in binding of the complex to the CAATT box. Interaction of the NF-YA subunit with the NF-YB/YC dimers occurs via the αC helices, which are well conserved in the *Plasmodium* ORP1 and ORP2 proteins. It has been determined that the NF-YC αC helix is especially important for interaction with NF-YA and other regulatory proteins[7]. The fact that this part of the protein is highly conserved in ORP2 may suggest that the ORP1/ORP2 dimer interacts with a third protein. BLAST searches have not identified an NF-YA orthologue in the malaria parasite and the fact that the ORP proteins are located in the oocyst capsule, a highly specialized structure of the parasite, argues against the role of the proteins in DNA binding. It is noteworthy that the αC helix of NF-YC also interacts with other regulatory proteins[15]. It is thus an interesting possibility that the ORP dimer may recruit a third factor, but the fact that the possible interaction of the two proteins in the mature oocyst capsule is a very transient event make the identification of other interacting proteins highly challenging. As far as we know, this is the first example of proteins containing well-conserved NF-YB and NF-YC domains residing outside the nucleus and having functions unrelated to DNA binding.

Importantly, the identification of ORP1 and ORP2 as essential proteins for mosquito transmission of malaria parasites provides new insights into the elusive oocyst capsule.

## Methods

**Ethics statement.** All work was carried out in full conformity with Greek and Italian regulations and laws on animal experiments. In Greece, these issues are covered by the Presidential Decree (160/91) and law (2015/92), which implement the directive 86/609/EEC from the European Union and the European Convention for the protection of vertebrate animals used for experimental and other scientific purposes, and the new legislation Presidential Decree 56/2013. The experiments were carried out in a certified animal facility licence (EL91-BIOexp-02) and the protocol has been approved by the FORTH Committee for Evaluation of Animal Procedures (6740/8/10/2014) and by the Prefecture of Crete (license number 27290, 15/12/2014). Animal experiments performed at the University of Perugia and at the Istituto Superiore di Sanità were approved by Italian Ministry of Health under the guidelines D.L. 116/92, which implement the directive 86/609/EEC from the European Union.

**Mosquito and parasite strains and methods.** *Anopheles gambiae* strain G3 mosquitoes were reared as described[16]. *P. berghei* ANKA 8417HP strain was used throughout the study[17] and was maintained in Theiler's Original mice or CD1 mice (Harlan Sprague Dawley, Inc.). Mice were male or female and ∼6–8 weeks when infected. Purification of blood stages were performed as described[18]. Asexual *Plasmodium* blood stages were purified at different time points after schizont culture synchronization[19]. Gametocytes were isolated from synchronized cultures 23 h after injection and removal of ring stage parasites by density gradient centrifugation using Nycodenz. Ookinetes were cultured *in vitro* according to ref. 20. Transfections were performed as described[21]. Mosquito infections were

carried out by offering *A. gambiae* strain G3 to mice infected with WT and mutant. The mice were matched to have a parasitemia of ∼5–10% and exflagellation was similar in each experiment (Supplementary Table 1). At least 21 midguts were counted in each group to ensure adequate statistical power[22]. Mosquito midguts and salivary glands were dissected at different time points p.b.f., homogenized in RPMI and sporozoites counted under the microscope. For pre-patent calculation midgut sporozoites were injected intravenously in 6–8 weeks C57BL/6 mice (Harlan Sprague Dawley, Inc.) and blood smears evaluated every day until day 15 post infection.

**Antibodies.** ORP1 was detected using a rabbit antiserum directed against an eight aa peptide of *P. falciparum* ORP1 (ref. 8; diluted 1:100 in IFA, 1:2,000 in western blot analysis). The monoclonal antibody recognizing CSP[23] (clone 3D11) was diluted 1:400. The antibodies directed against the capsule protein Cap380 (diluted 1:500) and the ookinete surface protein P28 clone 13.1 (diluted 1:400 in IFA) have been described[10,24]. An antibody recognizing mCherry was obtained from St John's Laboratories (catalogue number STJ34373). Secondary antibodies conjugated with Alexa Fluor were purchased from ThermoFisher Scientific (Alexa-Fluor 488 goat anti-mouse IgG catalogue number A11029, Alexa-Fluor 555 goat anti-mouse IgG catalogue number A21424, Alexa-Fluor 488 goat anti-rabbit IgG catalogue number A11034) and Cy3-conjugated donkey anti-rabbit IgG from Jackson Immunoresearch (catalogue number 711-165-152); they were all diluted 1:1,000. For western blotting, peroxidase-conjugated secondary antibodies were used; goat anti-mouse IgG, light chain and goat anti-rabbit IgG were obtained from Jackson Immunoresearch (catalogue numbers 115-035-174 and 111-035-045, respectively) used in 1:25,000 dilution. The tagged proteins expressed in *E. coli* were recognized by an anti-His-6 antibody (Abd Serotec, v MCA 1396) and an antibody recognizing the *S*-tag (St John's Laboratory, catalogue number STJ96919).

**Plasmid constructs for transfections.** The plasmids used for the generation of *orp1(−) cl1* and ORP::GFP transgenic parasites are derivatives from the pDEF-PbDHFR-SSU[25].

pOrp1(−) was generated by the addition of a 670 bp fragment of the 5′-flanking region (FR) in the ApaI–XhoI sites using primers 260S1for–260S2rev. A fragment of 607 bp amplified in the 3′-FR using primers 260D × 1for and 260D × 2rev was inserted in the sites KpnI–NotI. The vector was linearized using ApaI–NotI; transfected parasites were selected with pyrimethamine.

ORP1::GFP is expressed as an extra copy of the gene in the C/D ssu ribosomal RNA locus. The promoter region of 1,232 bp was amplified using primers TF5′for and TF5′rev, and inserted in the sites XhoI–XmaI. The fragment corresponding to the first 365 aa of the coding region were amplified by primers TF260for and TF260rev, and inserted using restriction enzymes XmaI–ApaI. The 482 bp ORP1 3′-FR region was amplified by TF3′ for and TF3′rev and inserted in the NheI and NotI restriction sites.

The pBAT plasmid[26] was used to generate the final constructs for the generation of ORP2::mCherry parasites, the *orp1 cl2*, *orp2(−)* and *orp1-hfd*⁻ mutants.

The transgenic parasite line expressing ORP2::mCherry was generated by introducing the coding region of mCherry in the endogenours locus by double crossover. To this end, the last 655 bp of the *orp2* coding region (amplified with primers 130340CODfor and 130340cod2rev) was fused in frame with the gene encoding mCherry in the pBAT vector using the enzymes SacII–SpeI. A 620 bp fragment of the 3′-FR of *orp2* (amplified with primers 130340tg-3′ for and 130340tg-3′rev) was inserted in the SphI–SacI sites. The last step is the insertion of a 751 bp homologous region for recombination just after the *orp2* 3′-FR region amplified by 130340tg-omfor and 130340tg-omrev in the ApaI–AatII sites. Finally, the plasmid was linearized using the AatII–SacII enzymes before transfection.

The *orp1(−) cl1* and *orp2(−)* parasites were generated by introducing the drug-selectable cassette linked to a high expression GFP cassette in pBAT[26] by double crossover. For the *orp1(−) cl2*, the gene was replaced via a double crossover, targeting the same regions as in *orp1(−) cl1*, integrating the recyclable

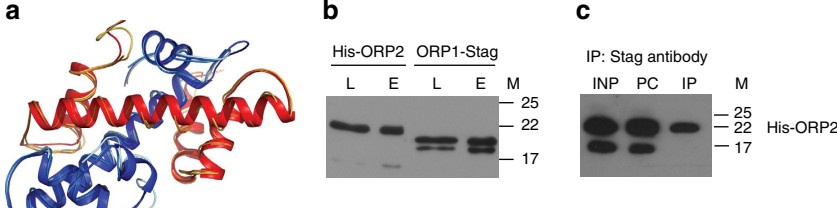

**a** ORP2 aa 1–875    mCherry

**b**

|  | mCherry | ORP1 | mCherry/ORP1 |
|---|---|---|---|
| Day 7 | | | |
| Day 11 | | | |

**c**

|  | mCherry | Cap380 | mCherry/Cap380 |
|---|---|---|---|
| Day 13 | | | |

**d**

**Figure 3 | ORP2 is re-localized from the cytoplasm to the oocyst capsule concomitantly with oocyst rupture.** (**a**) ORP2::mCherry construct. (**b**) Double labelling of ORP2::mCherry (red) and ORP1 (green) in oocysts 7 and 10 days p.b.f. Sporozoites have not yet formed. ORP2 is localized in the cytoplasm. Scale bars, 15 and 20 µm. (**c**) A mature oocyst at day 13 p.b.f. ORP2 is detected at the periphery partly overlapping with the Cap380 marker of the oocyst capsule. Scale bar, 10 µm. (**d**) Higher magnification of the oocyst in **c** reveals ORP2 (red) localization close to the capsule highlighted with Cap380 (green). Scale bar, 2.5 µm.

**a**    **b**    His-ORP2    ORP1-Stag    **c**    IP: Stag antibody

**Figure 4 | ORP1 and ORP2 HFDs form a dimer *in vitro*.** (**a**) Structural modelling of ORP1/ORP2 dimer together with human NF-YB and NF-YC chains used as templates. Thick ribbons show ORP1 in red and ORP2 in blue. The thin orange and middle blue ribbons present the template chains B and C of 1N1J.pdb, respectively, and the thin yellow and light blue ribbons the template chains B and C of 4AWL.pdb, respectively. (**b**) Pull-down assay of His-ORP2 with Ni-NTA beads reveal binding of ORP1. Gene sequences encoding peptides corresponding to each HFD with short flanking sequence were cloned in the pET-Duet-1 vector and the proteins expressed together in *E. coli*. ORP1 was tagged C-terminally to the *S*-tag and ORP2 N-terminally to a poly-His tag. The western blotting was loaded with cleared lysate (lanes L) and eluate from the Ni-NTA bead (lane E) and the proteins visualized by commercial antibodies directed against the tags. M, molecular weight marker in kDa. (**c**) Co-immunoprecipitation using the antibody against the *S*-tag to capture the complex from bacterial lysate. His-ORP2 was detected in the immuno-precipitated fraction. INP, input, bacterial lysate; PC, pre-cleaned lysate; IP, precipitated fraction; M, molecular weight marker in kDa.

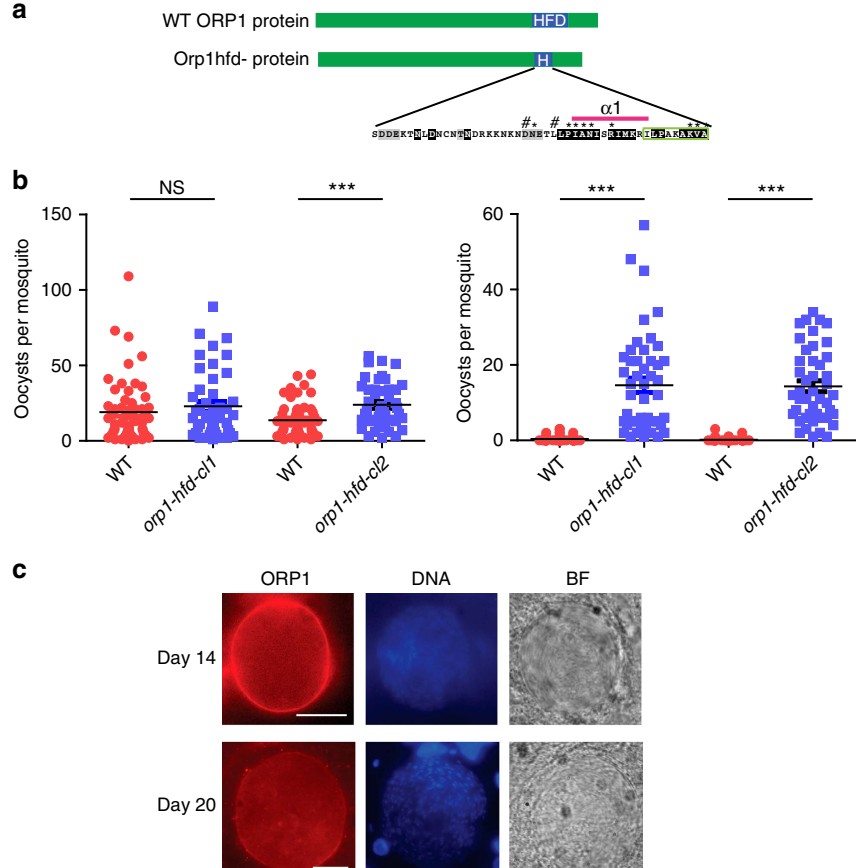

**Figure 5 | The HFD is necessary for ORP1 function. (a)** Schematic depiction of the construct encoding ORP1 in which the α2 to αC helices (aa 797–860) of the HFD were deleted leaving the epitope for the ORP1 antibody intact (green box). **(b)** Oocyst load in mosquitoes infected with *orp1-hfd⁻* compared with WT at day 11 (left) and day 21 p.b.f. (right). Pooled data from two independent experiments of each clone. NS, nonsignificant, ***$P < 0.0001$, Mann–Whitney test. The complete data set is presented in Supplementary Fig. 9 and Supplementary Table 1. Error bars denote s.e.m. **(c)** ORP1 lacking the HFD is localized at the oocyst capsule at both day 14 and 20 p.b.f. Sporozoites are visible inside the oocysts. Nuclei stained with 4,6-diamidino-2-phenylindole (DAPI; blue) and the bright field (BF) image. Scale bars, 20 µm.

hDHFR-yFcu cassette, which encodes anti-folate resistance. GFP expression was regulated by the PbHSP70 promoter. After transfection, $5 \times 10^5$ GFP-fluorescent red blood cells were sorted and intravenously injected into two CD1 mice and kept under pyrimethamine selection. When parasitaemia was established, sorting and new infections under selection were repeated two more times.

The *orp2* construct comprised a 974 bp PCR fragment of the 5′-FR of the gene amplified with primers 130340S1 and 130340S2, inserted in the restriction sites SacII–SpeI and an 827 bp fragment of the 3′-FR amplified with primers 130340D1 and 130340D2 inserted in the sites ApaI–XhoI. The plasmid was digested with the enzyme SalI before transfection. The two clones were derived from independent transfections.

The deletion corresponding to aa 797–860 of the HFD of ORP1 was generated by introducing a nested PCR fragment. This included the last 890 bp of the first exon and 25 bp encoding c-myc (amplified with primers ORP1-(targSIN-myc)-for and ORP1(targSIN-myc)-revand), and a fragment containing 25 bp encoding c-myc in frame with the last 270 bp of the coding region followed by the stop codon and 595 bp of the 3′-FR (amplified by primers ORP1(myc-fine)-for and ORP1(myc-fine)-rev). The fragment was inserted in pBATSIL6 using the restriction sites SacII–SacI. A 597 bp fragment, amplified with primers MutORP1-targDX-for and MutORP1-targDX-rev, corresponding to the 3′-FR of *orp1* was inserted in the restriction sites MluI–AatII. The final plasmid was linearized with PvuI before transfection. The two clones were derived from independent transfections.

All primer sequences are available on request.

**Immunofluorescence analysis.** Midguts were dissected at different days after the blood meal and fixed for 1 h in 4% paraformaldehyde/1% saponin in PBS. In the following, all incubations were carried out for 1 h at room temperature. After three washes in PBS/1% saponin (PBSS), parasites were blocked with 5% normal goat serum (NGS)/PBSS, incubated with the primary antibody diluted in PBSS, washed several times in PBS after which the secondary antibodies were added (1:800 dilution). Cell nuclei were labelled with 4,6-diamidino-2-phenylindole. Ookinetes

and sporozoites were fixed as above and permeabilized with 0.1% Triton X-100 in PBS, followed by incubation with the primary antibodies as described. The specificity of the immune sera was checked in parallel using pre-immune sera and controls were included to exclude nonspecific binding of the secondary antibodies; they were all negative. Samples were viewed using a Zeiss Axioskop 2 plus microscope and, in some cases, analysed using Zeiss LSM 510 confocal laser scanning microscope. Images were further processed using Image J.

**Structural modelling.** The model was built based on two structures of the human NF-YB and NF-YC dimer. The chains C of 4AWL.pdb (3 Å)[7] and 1N1J.pdb (1.6 Å)[15] were structural aligned (superimposed) in the Swiss-PdbViewer. The raw sequence of ORP2 (amino acid residues 27–109) was aligned manually to the superimposed chains C with two different positions of the 3 residue insert (as DKI or as NSQ). Two modelling requests with different positions of the insert were submitted to the Swissmodelserver (http://swissmodel.expasy.org/) and models with bad geometry and high energy were received. As the geometry of the received second model was better, this was chosen for further modelling. Next, ORP1 (amino acids 769–860) was loaded as a raw sequence to the superimposed structures of 4AWL.pdb and 1N1J.pdb, whereby the new modelled chain C was also loaded. Another modelling request was submitted and the model of both chains B and C was received.

Energy minimization with cns_solve of the dimer was performed with the following steps. Setting up the structure file mtf, adding hydrogens in calculated positions, calculating the backbone intrachain hydrogen bonds and the six interchain backbone hydrogen bonds. As this programme cannot calculate hydrogen bonds involving side chains, the seven hydrogen bonds, which should exist in analogy to the two structures, were added in the input file manually. The hydrogen definition file was used as restraints input file. The dimer model was energy minimized (*in vacuo*, 1,200 cycles of conjugate gradient minimization with hydrogen bond restraints, non-bonded cutoff 13 Å). In addition, the six backbone–backbone and the seven hydrogen bonds involving side chains and

seven other possible hydrogen bonds were built during the energy minimization process.

**Pull-down assay and co-immunoprecipitation.** The two proteins were expressed from the pET-Duet-1 vector (Novagen). A fragment corresponding to aa 7–156 of ORP2 was amplified from genomic DNA and inserted in the plasmid using restriction sites BamHI and HindIII. Thus, the recombinant protein was N-terminally tagged with poly-histidine. Next, a PCR fragment was amplified from complementary DNA derived from mixed blood stages corresponding to ORP1 aa 750–872. It was inserted in the ORP2-pET-Duet-1 plasmid using restriction sites NdeI and XhoI, allowing read through to the *S*-tag sequence of the vector. In a control construct, the *orp1* fragment only was cloned in pET-Duet-1 using the same restriction sites. *E. coli* strain BL21(DE3) containing the plasmid pMICO[27] was transformed with the plasmids. Cultures were harvested 2 h after induction of expression with isopropyl-β-D-thiogalactoside of the cells containing the ORP1/ORP2-pET Duet-1, while at 4 h of the ORP1-pET-Duet-1 plasmid cells. The time was chosen for maximal expression of the proteins. Pelleted cells were lysed in 500 mM NaCl, 150 mM Tris pH 8.0, 10% glycerol, 10 mM immidazole (Sigma-Aldrich) containing 1 mM phenylmethylsulfonyl fluoride (PMSF) by sonication on ice five times 30 s. All of the following steps were carried out at 4 °C. The lysate was cleared by centrifugation at 20,000 g for 30 min and then mixed with 100 μl Ni-NTA (Qiagen) beads previously equilibrated with the lysis buffer. The mixture was incubated overnight with constant shaking. The Ni-NTA beads were then collected by centrifugation at 800 g for 5 min and washed once with the lysis buffer, followed by three washes with 300 mM NaCl, 150 mM Tris pH 8.0, 10% glycerol and 10 mM imidazole. Elution was carried out in 300 mM NaCl, 150 mM Tris pH 8.0, 10% glycerol and 300 mM imidazole. Samples of the cleared lysate and the elution was loaded on 15% SDS–PAGE gels and processed for western blottings.

Co-immunoprecipitation of the tagged ORP HFD peptides was carried out by incubating Protein G sepharose beads with the antibody against the *S*-tag, followed by incubation with the cleared bacterial lysate prepared as above, but lysed in 300 mM NaCl, 50 mM Tris pH 7.4, 1% Triton X-100 and 1 mM PMSF. After extensive washing in wash buffer (300 mM NaCl, 50 mM Tris pH 7.4, 0.1% Triton X-100 and 1 mM PMSF), the beads were re-suspended in sample buffer and loaded on SDS–PAGE. The His-6 antibody was used to recognize the ORP2 HFD peptide.

**Statistics.** Statistics analysis was carried out on GraphPad Prism Software. Infection intensity was compared between strains using the Mann-Whitney test (two-tailed).

**Data availability.** The data that support the findings of this study are available from the corresponding author upon request.

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

## Acknowledgements

We thank Professor J. Papamatheakis for comments on the manuscript, and Mr Dimitris Malisiovas, Ms Aurélie Fougere, Dr Valentina Tirelli and Dr Massimo Sanchez for assistance. C.C. was a recipient of a fellowship from the I-MOVE Fellowship Programme—PCOFUND-GA-2010-267332 (Marie Curie). The project was supported by the BIOSYS research project, Action KRIPIS, project number MIS-448301 (2013SE01380036) funded by the General Secretariat for Research and Technology, Ministry of Education, Greece, and the European Regional Development Fund, the Italian FLAGSHIP 'InterOmics' project (PB.P05) funded by MIUR, and coordinated by the CNR and by Fundação de Amparo à Pesquisa de São Paulo (FAPESP) (Process 2011/51295-5). In the initial phase of the project, C.C. was funded by a fellowship from FAPESP (Process 2011/236267).

## Author contributions

C.C., I.S.-K., M.P., R.S. and C.R.S.G. designed the study. C.C., R.G., T.P., L.P. and G.P. performed most of the experiments. C.C., R.G. and T.P. were responsible for construction and validation of the mutants. C.C. did most of the phenotypic analysis with assistance from G.P. on the sporozoite data. R.G. did the structural model. V.V.-M. and L.S. carried out the bacterial expression and peptide interactions. C.C., I.S.-K., M.P. and R.S. analysed the data. I.S.-K. wrote the manuscript with participation of C.C., M.P., R.S. and C.R.S.G.

## Additional information

**Competing financial interests:** The authors declare no competing financial interests.

**Publisher's note**: 

