## [Peer Review File · Nature Communications]

Reviewers' comments:

Reviewer #1 (Remarks to the Author):

The paper has identified two proteins (which should be identified by their gene number from the very outset), which from their structure might be expected to be localized to the nucleus, in the extracellular capsule of the oocyst stage of the malaria parasite life cycle. This is a novel observation of interest to both malariologists and cell biologists.

The work is in general well conducted/presented and the data largely valid. One area where the methodology is not well described and the experiments 'unconventional', relates to the comparative infectivity of the GM parasites vs the controls. There is no evidence that the gametocyte challenge for the infections has been made equivalent to the WT, and apparently raw data has been pooled between expts, this is not correct. It is essential to compare WT/GM within carefully-paired expts, and then pool the relative infectivities. The only conclusion affected is the statement that the GM lines are more infectious than the WT. One point of interest not remarked upon by the authors - why if there is an impaired escape-phenotype does the oocyst number in the GM lines fall by 50%?

On pages 12/13 the DAPI images are not of high enough clarity to support the obvious interpretation that it is nuclei that are stained.

Could the authors provide a more considered analysis as to why the co-localization studies of ORP/CSP/380 show only partial superimposition, is it capsule (internal) basal lamina (external) differences? or perhaps their instrumentation is not localizing the different wavelengths equivalently (DIC/nonDIC optics?).

Whilst the possible structural analysis of the proteins is eminently logical, the summary paragraph does take the interpretation of the structure/function relationship way beyond the proven data. This is a nice piece of work which, subject to the reanalysis of the infection data, perhaps could be 'toned down' on the structural biology aspects and will then provide a very useful new contribution to the literature.

Invite a re-submission

Reviewer #2 (Remarks to the Author):

The manuscript entitled "Release of Plasmodium sporozoites requires proteins with histone fold dimerization domains" describes an original finding that isogenic Plasmodium berghei parasites lacking proteins called ORP1 or ORP2 cannot egress from oocysts in the mosquito midgut. This is an interesting observation. However the number of experimental replicates in Table 1 and 2, and the number of knockout clones studied, is inadequate to conclude anything. In the absence of genetic complementation, at least two individual knockout clones are required to conclusively demonstrate the importance of these proteins. Furthermore, several controls are missing from key experiments (see below). Finally, there is no mechanistic insight provided for how these proteins function in the phenotypes described. In other organisms, these two proteins bind DNA via a third protein that, in Plasmodium, is suggested to be absent. So what do ORP1 and ORP2 do? There is no discussion to help answer these questions. The important claim that ORP1 binds ORP2 directly by in silico modeling is not supported by any experimental evidence which is a big weakness in the current manuscript.

Key points:

Table 1 shows two independent experiments but they are not direct repeats, as the day of dissection is different and the number of WT sporozoites varies greatly. At least a third experimental repeat is required where the conditions were exactly repeated.

Table 2 presents duplicate experiments for inoculation by mosquito bite. Experiment 3 is not in fact a

third repeat but is a different experiment where sporozoites were injected into the tail veins. As such there is only one experiment presented for iv inoculation and it needs repeating such that n=2. The information provided from these experiments is interesting but incomplete. If the mutants cannot establish a patent infection, where do they arrest? Do they infect mouse livers? An early and late time point for liver infection in mice by qRT-PCR would be important, as this would address: i) whether the mice became infected early, and ii) the timing at which the mutant(s) arrests, providing important information about the phenotypes involved during mammalian infection.

How many knockout clones for each ORP mutant were studied? The gold standard is that at least 2 individual clones are investigated to mitigate the possibility of other genes contributing to the phenotypes observed (see PMID 20667784). Indeed, there is no complementation data provided to unequivocally link the phenotypes observed to the genes they are being ascribed to. At the very least, 2 knockout clones are required or the authors should complement their mutants.

Figure 1d, e. The authors declare ORP1 and ORP2 are not required for asexual, gametocyte, ookinete or sporozoite development but that data presented on for each of these claims is either entirely absent or very sparse. sporozoite development is light on. The rate of parasite growth in asexual stages, the number of mature gametocytes, the number of ookinetes formed, and the number of sporozoites liberated by mechanical lysis of oocysts, is required compared to WT. Multiple KO clones tested in duplicate experiments are necessary to substantiate the important claims.

A key question for lack of salivary gland sporozoites and also subsequent infection in mice is whether the mutant sporozoites are motile. The authors should measure motility of mutants compared to WT in experimental replicates.

The ORP1 and ORP2 antibodies require substantial validation. Antibody specificity should be presented in full-length immunoblots from ookinetes, oocysts or sporozoites so as to allow the reader to determine the specificity of the signal in the micrographs shown. The blot shown in Figure S3 is not full length and lacks the isogenic deletion mutant controls. Similarly, the authors should show immunofluorescence images using the antibodies against knock out parasites to demonstrate their specificity in microscopy.

Structural modelling is interesting but not proof of a real interaction. For publication in Nature Communications, the authors might be expected to experimentally prove that the two proteins interact. This could be demonstrated either by immunoprecipitation of native or tagged proteins from parasite lysates or through recombinant expression and co-IP.

Minor points:

Why are the authors suggesting that the salivary gland sporozoites recovered are contaminants? Is there clear evidence of this? If so, it should be provided. If it is just speculative then the statement should be removed from results and described in the discussion.

Page 4: regarding the 7 peptide region used for the antibody. The box is presented in Figure 1b not 1a as stated.

Figure 2. Double labeling does show that ORP1 partially co-localizes with Cap380. However a substantial signal is seen internal to the oocyst and this should also be described. For co-localization, multiple cells should be used (from z-stacks if possible) and a Pearson coefficient provided. It is unclear what the reference to laminin is implying? In sporozoites, what is the proposed reason for the extracellular staining? How often was then seen? Is it really a specific signal?

In Figure 4e, it is difficult to see the sporozoites indicated in the text of the figure legend. It would be helpful to provide an arrow or similar. Also can the authors quantify the number of sporozoites in the oocysts of the HFP-mutant parasites relative to WT following mechanical lysis of oocysts?

It must be the format of the manuscript submitted, but there is no discussion. This is a shame as there are many questions left unanswered.

Answer to Reviewers' comments:

Reviewer #1 (Remarks to the Author):

The paper has identified two proteins (which should be identified by their gene number from the very outset), which from their structure might be expected to be localized to the nucleus, in the extracellular capsule of the oocyst stage of the malaria parasite life cycle.

The gene numbers are now added in the Introduction.

This is a novel observation of interest to both malariologists and cell biologists.

The work is in general well conducted/presented and the data largely valid. One area where the methodology is not well described and the experiments 'unconventional', relates to the comparative infectivity of the GM parasites vs the controls. There is no evidence that the gametocyte challenge for the infections has been made equivalent to the WT, and apparently raw data has been pooled between expts, this is not correct. It is essential to compare WT/GM within carefully-paired expts, and then pool the relative infectivities.

We follow the standard in mosquito feeding experiments on *P. berghei* infected mice using mice infected with WT and mutant lines with similar parasitemia and exflagellation in vitro. The mice had a parasitemia of roughly 10% and exflagellation was similar in each experiment. In the new version of the paper we carried out four experiments. In each experiment one WT and one clone of each mutant were included (2 experiments/clone) and fed to the same population of mosquitoes. For clarity we chose to present the data pooled for each mutant clone in the main manuscript. However, each experiment is also presented as graphs with oocyst counts from each experiment (Supplementary Fig. 4 and 9) and as a table (Supplementary Table 1) with the complete dataset (including parasitemia, number of exflagellations, ookinete conversion and P-values for oocyst counts) as well as number of mosquitoes scored.

The only conclusion affected is the statement that the GM lines are more infectious than the WT. One point of interest not remarked upon by the authors - why if there is an impaired escape-phenotype does the oocyst number in the GM lines fall by 50%?

The escape phenotype is based on 1) counting the oocyst at 21 days pbf 2) immunolabeling using the CAP380 antibody which labeled intact oocyst capsule at 20 day in the mutant, while very few and degenerated oocysts were detected in the WT at the same time. 3) the fact that the sporozoite were motile but did not invade the salivary glands. For the comment on the 50% reduction we believe the reviewer refers to the difference between the oocyst counts at 12 and 20 days pbf (in the new manuscript day 11 and day 21). In some cases, as can be seen in Supplementary Table 1, there is a decrease in the average number of oocysts between the time points, but in other cases not. Such fluctuations may be due to chance as oocyst counts vary between individual mosquitoes.

On pages 12/13 the DAPI images are not of high enough clarity to support the obvious interpretation that it is nuclei that are stained.

We do not know to which figure this refers to. It is difficult to visualize the individual nuclei in the tightly packed mature oocyst.

Could the authors provide a more considered analysis as to why the co-localization studies of ORP/CSP/380 show only partial superimposition, is it capsule (internal) basal lamina (external) differences? or perhaps their instrumentation is not localizing the different wavelengths equivalently (DIC/nonDIC optics?).

We believe the reviewer refers to Fig. 2 c, d and Fig. 3 d. In Fig. 2c we used a parasite expressing a GFP fusion of ORP1 in double labeling experiment with CAP380. This figure has now been replaced with an image where we used confocal microscopy. In this figure the two colours overlap, but the signal of GFP was weak precluding detailed analysis of overlapping. In Fig. 2d we used the CSP in double labeling experiments with the ORP1 antibody to verify that ORP1 is localized outside the cell membrane, which suggests that the protein is found in the capsule. CAP380 and ORP1 antibodies are both of rabbit origin which prevented us from doing a direct co-labeling experiment. In Fig. 3d the overlap of ORP2::mCherry and CAP380 is partial. We believe this is reflecting the fact that the ORP2 protein is transiently translocated to the capsule just before rupture. We only detected a few in each midgut with this pattern of staining, all of which contained mature sporozoites.

Whilst the possible structural analysis of the proteins is eminently logical, the summary paragraph does take the interpretation of the structure/function relationship way beyond the proven data

This is a nice piece of work which, subject to the reanalysis of the infection data, perhaps could be 'toned down' on the structural biology aspects and will then provide a very useful new contribution to the literature.

Invite a re-submission

Reviewer #2 (Remarks to the Author):

The manuscript entitled "Release of Plasmodium sporozoites requires proteins with histone fold dimerization domains" describes an original finding that isogenic Plasmodium berghei parasites lacking proteins called ORP1 or ORP2 cannot egress from oocysts in the mosquito midgut. This is an interesting observation.

However the number of experimental replicates in Table 1 and 2, and the number of knockout clones studied, is inadequate to conclude anything. In the absence of genetic complementation, at least two individual knockout clones are required to conclusively demonstrate the importance of these proteins.

As discussed above we provide a new series of experiments where two clones of each mutant were studied, these have now been added to Table 1. We have also added a replicate to Table 2.

Furthermore, several controls are missing from key experiments (see below). Finally, there is no mechanistic insight provided for how these proteins function in the phenotypes described. In other organisms, these two proteins bind DNA via a third protein that, in *Plasmodium*, is suggested to be absent. So what do ORP1 and ORP2 do? There is no discussion to help answer these questions. The important claim that ORP1 binds ORP2 directly by in silico modeling is not supported by any experimental evidence which is a big weakness in the current manuscript.

Key points:

Table 1 shows two independent experiments but they are not direct repeats, as the day of dissection is different and the number of WT sporozoites varies greatly. At least a third experimental repeat is required where the conditions were exactly repeated.

Table 1 now contains 3 experiments, the first two being exact repeats (each represent pooled data of two experiments of each clone and the raw data is found in Supplementary Table 1).

Table 2 presents duplicate experiments for inoculation by mosquito bite. Experiment 3 is not in fact a third repeat but is a different experiment where sporozoites were injected into the tail veins. As such there is only one experiment presented for iv inoculation and it needs repeating such that $n=2$.

We have repeated the i.v. experiments using the second clones of the disruption mutants. The results are similar.

The information provided from these experiments is interesting but incomplete. If the mutants cannot establish a patent infection, where do they arrest? Do they infect mouse livers? An early and late time point for liver infection in mice by qRT-PCR would be important, as this would address: i) whether the mice became infected early, and ii) the timing at which the mutant(s) arrests, providing important information about the phenotypes involved during mammalian infection.

We attempted experiments addressing the reviewers concern in this matter. To determine if the sporozoites can infect liver cells in vitro infections were attempted. However, the sporozoites in this case come from the midgut and these must be squeezed to liberate the sporozoites. Thus a high number of bacteria were also released together with the parasites, and the liver cell culture was heavily contaminated (despite the addition of antibiotics) which precluded any conclusions. We also carried out PCR from livers of infected mice, but no fragment was amplified from the mutants, while a fragment was obtained from the WT, as expected. This suggests that the block in infection is at an early stage although we cannot pinpoint the exact stage. We have added this information to the manuscript (Lines 101-103).

How many knockout clones for each ORP mutant were studied? The gold standard is that at least 2 individual clones are investigated to mitigate the possibility of other genes contributing to the phenotypes observed (see PMID 20667784). Indeed, there is no

complementation data provided to unequivocally link the phenotypes observed to the genes they are being ascribed to. At the very least, 2 knockout clones are required or the authors should complement their mutants.

We have isolated an independent second clone for each mutant. In one case, *orp1(-)* a completely new construct was generated for the second clone. As discussed above we now present data from four separate experiments, two for each clone. The results are similar to our previous data.

Figure 1d, e. The authors declare ORP1 and ORP2 are not required for asexual, gametocyte, ookinete or sporozoite development but that data presented on for each of these claims is either entirely absent or very sparse. sporozoite development is light on. The rate of parasite growth in asexual stages, the number of mature gametocytes, the number of ookinetes formed, and the number of sporozoites liberated by mechanical lysis of oocysts, is required compared to WT. Multiple KO clones tested in duplicate experiments are necessary to substantiate the important claims.

We now present the complete dataset of two clones of each mutant in Supplementary Table 1, and in Supplementary Fig. 3 (growth rates).

A key question for lack of salivary gland sporozoites and also subsequent infection in mice is whether the mutant sporozoites are motile. The authors should measure motility of mutants compared to WT in experimental replicates.

We have added Supplementary Fig. 5 showing typical trails of moving sporozoites from midgut sporozoites obtained from mechanically ruptured oocysts 17 d pbf of the two disruption mutants.

The ORP1 and ORP2 antibodies require substantial validation. Antibody specificity should be presented in full-length immunoblots from ookinetes, oocysts or sporozoites so as to allow the reader to determine the specificity of the signal in the micrographs shown. The blot shown in Figure S3 is not full length and lacks the isogenic deletion mutant controls. Similarly, the authors should show immunofluorescence images using the antibodies against knock out parasites to demonstrate their specificity in microscopy.

The ORP1 antibody validation was shown in the previous manuscript (Supplementary Fig. 4a) as immunofluorescence assay of WT and mutant oocysts. We have now added a Western blot comparing WT and mutant ookinetes (New Supplementary Fig. 6b). We also show the whole Western blot of the different life stages of WT parasites probed with the ORP1 antibody (Supplementary Fig. 6c), as requested by the reviewer. The ORP2 protein was detected using a commercial m-Cherry antibody. In these experiments a transgenic parasite where the endogenous ORP2 is fused to mCherry was used.

Structural modelling is interesting but not proof of a real interaction. For publication in Nature Communications, the authors might be expected to experimentally prove that the

two proteins interact. This could be demonstrated either by immunoprecipitation of native or tagged proteins from parasite lysates or through recombinant expression and co-IP.

We believe that combining the facts that ORP1 and ORP2 knockouts gave the same phenotype, and the small deletion of the C-terminal part of the ORP1 HFD gave the same phenotype as the complete deletion of the *orp1* coding region is strong evidence that 1) the two proteins interact, directly or indirectly 2) the HFD has a critical role in the function of the protein. Together with the fact that the highly conserved HFDs are well-documented domains for protein interaction suggest that the two proteins do interact directly via the HFD. To strengthen this notion we have expressed the two ORP HFDs in bacteria, and we show that these peptides interact in vitro, using a pull-down assay and co-immunoprecipitation (New Fig. 4).

Minor points:

Why are the authors suggesting that the salivary gland sporozoites recovered are contaminants? Is there clear evidence of this? If so, it should be provided. If it is just speculative then the statement should be removed from results and described in the discussion.

We base this on the fact that when dissections were carried out carefully we did not recover sporozoites in the salivary glands, and we have removed this experiment from Table 1 for this reason (Table 1).

Page 4: regarding the 7 peptide region used for the antibody. The box is presented in Figure 1b not 1a as stated.

Corrected

Figure 2. Double labeling does show that ORP1 partially co-localizes with Cap380. However a substantial signal is seen internal to the oocyst and this should also be described. For co-localization, multiple cells should be used (from z-stacks if possible) and a Pearson coefficient provided.

We have replaced these pictures in Fig. 2c with a new confocal image, which we hope will be easier to interpret. We appreciate that the co-localization experiments need careful analysis to be meaningful. The GFP label in Fig. 2c was weak and this impeded analysis such as Pearson correlation coefficient. However, together with the result in Fig. 2d where the ORP1 signal is clearly localized outside CSP strengthens the notion that ORP is a capsule protein.

It is unclear what the reference to laminin is implying?

The oocyst capsule has been suggested to contain laminin from the mosquito. We removed this sentence to avoid confusion.

In sporozoites, what is the proposed reason for the extracellular staining? How often was then seen? Is it really a specific signal?

We believe the reviewer discusses Fig. 2c. We do not believe this is a specific signal and we replaced the picture with a more typical sporozoite, where ORP1 is only seen in the cytoplasm.

In Figure 4e, it is difficult to see the sporozoites indicated in the text of the figure legend. It would be helpful to provide an arrow or similar. Also can the authors quantify the number of sporozoites in the oocysts of the HFP-mutant parasites relative to WT following mechanical lysis of oocysts?

The oocyst is filled with tightly packed sporozoites so it is not possible to indicate a single sporozoite in the figure. The number of the sporozoites after mechanical rupture is now included in Supplementary Table 1.

It must be the format of the manuscript submitted, but there is no discussion. This is a shame as there are many questions left unanswered.

The manuscript has been re-formatted and a separate Discussion has been added.

REVIEWERS' COMMENTS:

Reviewer #2 (Remarks to the Author):

The authors have addressed all of my major concerns and are to be congratulated on significantly strengthening the paper. I recommend acceptance once:

1. the qPCR data on lack of liver-stage development in mice is included in the manuscript including full materials and methods. I acknowledge that this is not the main finding in the paper but it is still interesting and should be included. Currently it is unclear how many mice were infected and what the PCR "fragment" to measure the infection actually is. The authors should use a primer pair to the parasite (e.g. Pb18S) and another pair to a mouse gene that is expressed in hepatocytes (e.g. mHPRT or similar). Please ensure sufficient replicates of this experiment are included.
2. Figure 2a: in the version of the Figure that I see, the "merge" panel appears to contain more intensity than the individual DNA and P28 panels, which requires correction.

Answer to reviewers comments

Reviewer #2 (Remarks to the Author):

The authors have addressed all of my major concerns and are to be congratulated on significantly strengthening the paper. I recommend acceptance once:

1. the qPCR data on lack of liver-stage development in mice is included in the manuscript including full materials and methods. I acknowledge that this is not the main finding in the paper but it is still interesting and should be included. Currently it is unclear how many mice were infected and what the PCR “fragment” to measure the infection actually is. The authors should use a primer pair to the parasite (e.g. Pb18S) and another pair to a mouse gene that is expressed in hepatocytes (e.g. mHPRT or similar). Please ensure sufficient replicates of this experiment are included.

We have now inserted a new Supplementary Fig. 5b, where RT-PCR data from dissected livers of the *orp1(-)* and *orp2(-)* injected mice (3 animals in each case) compared to WT using Pb18S primers specific for the parasite. Primers recognizing the mouse *gapdh* gene were used as quality control of the cDNA. The experimental details are provided in the Supplementary Methods section.

2. Figure 2a: in the version of the Figure that I see, the “merge” panel appears to contain more intensity than the individual DNA and P28 panels, which requires correction.

This has been corrected.

The comments of Reviewer 1 have been corrected and answered in the text.